# Unveiling the Hidden Perils: A Comprehensive Review of Fungal Infections in Inflatable Penile Prosthesis Surgery

**DOI:** 10.3390/jpm14060644

**Published:** 2024-06-16

**Authors:** Anastasios Natsos, Vasileios Tatanis, Alexandra Lekkou, Stavros Kontogiannis, Athanasios Vagionis, Theodoros Spinos, Angelis Peteinaris, Mohammed Obaidat, Konstantinos Pagonis, Panagiotis Kallidonis, Evangelos Liatsikos, Petros Drettas

**Affiliations:** 1Department of Urology, University of Patras Hospital, 26504 Patras, Greece; stavroskontogiannis@gmail.com (S.K.); thanos_vagionis@hotmail.gr (A.V.); thspinos@otenet.gr (T.S.); peteinarisaggelis@gmail.com (A.P.); kasious.klay@gmail.com (M.O.); pagonisk7@gmail.com (K.P.); pkallidonis@yahoo.com (P.K.); liatsikos@yahoo.com (E.L.); drettas@gmail.com (P.D.); 2Department of Infectious Diseases, University General Hospital of Patras, 26504 Patras, Greece; alekkou@yahoo.gr; 3Department of Urology, Medical University of Vienna, 1090 Vienna, Austria

**Keywords:** fungal infections, penile implants, penile prosthesis, diabetes mellitus, guidelines, sexual dysfunction

## Abstract

Inflatable penile prosthesis (IPP) surgery is an effective treatment for erectile dysfunction (ED), but infections pose a significant threat to its success. Current guidelines lack antifungal recommendations, despite rising fungal infection rates post-IPP surgery. This review examines epidemiology, risk factors (including diabetes mellitus, immunosuppression, and obesity), and pathogenesis, highlighting the role of biofilm formation in device contamination. Clinical manifestations vary from acute to delayed, with fungal biofilms presenting challenges in diagnosis. Prophylactic strategies, including broad-spectrum antibiotics and antifungals, are crucial, with evidence suggesting a 92% reduction in infections. With fungal infections showing lower salvage rates, management involves culture-guided treatment, irrigation, and oral antibiotics. Future research aims to understand biofilm mechanisms and develop biomaterials to reduce infection rates. Implementing antifungal therapy, along with standard practices like the no-touch technique and antibiotic dips, is crucial in preventing IPP infections.

## 1. Introduction and Background

Inflatable penile prosthesis (IPP) surgery remains one of the cornerstones in the management of erectile dysfunction (ED). However, infections pose a significant threat to compromising implant success, with detrimental effects on the patient’s quality of life if salvage therapy is not feasible. While established guidelines from the European Association of Urology (EAU) and American Urologic Association (AUA) address infection prophylaxis in IPP surgery, they currently do not include antifungal agents [1,2]. Interestingly, their guidance deviates even on basic topics, such as length of antibiotic coverage, with the EAU recommending prolonged antibiotic use of over 24 h and the AUA up to 24 h. The omission of antifungal prophylaxis is increasingly challenged by recent research highlighting the rising incidence of fungal infections after IPP [3]. Added to this, critics argue that existing guidelines rely on data extrapolated from other prosthetic surgeries, and not specifically IPP [4]. Fungal infections can be insidious, often presenting with delayed symptoms that jeopardize implant functionality. They can lead to devastating consequences, ranging from implant failure to life-threatening sepsis [5]. The complex interplay between patient immune status, surgical technique, and fungal virulence necessitates a multi-faceted approach to the prevention, diagnosis, and treatment of these infections. This review article comprehensively explores fungal complications in IPP surgery regarding epidemiology, pathogenesis, clinical signs and symptoms, diagnostic methods, and current treatment strategies. By raising awareness of this critical but often overlooked complication, we aim to equip the urologic surgeon with the knowledge and tools to optimize patient outcomes.

## 2. Review

### 2.1. Epidemiology

Generally, the infection rate after primary IPP surgery is found to be less than 1–3%; however, this depends on modifiable risk factors (i.e., obesity) [6]. Also, the incidence of infections may increase to 10% in case of revision surgery, and even 18% in concomitant diabetes mellitus (DM) [5]. Fungal infections following penile implant surgery are relatively uncommon compared to bacterial infections. However, the exact incidence varies depending on various factors, such as patient population, surgical techniques, preoperative preparation, and postoperative care protocols. Studies have reported that fungal infections account for approximately 5% of all infections [7], but a large, multicenter study by Gross et al. found that 11.1% of cultures yielded a Candida species infection [8]. It should be noted that in the latter study, only 1 of 227 culture-positive patients received fluconazole. Chandrapal et al. showed that in their removal and replacement protocol for salvage IPP surgery, fungi accounted for up to 33% of all positive cultures found [9].

### 2.2. Risk Factors

Various risk factors have been identified in developing IPP infection. Some of those risk factors are associated with an increased predilection for fungal infections. DM is known to increase overall IPP infection with an odds ratio (OR) of 2.48 (95% CI 1.38–4.47) [6]. Regarding fungal infections, 69% of patients suffering from fungal IPP infections had DM [10]. Although newer evidence suggests that a cut-off of 8.5% of HbA1c levels is statistically significant in prohibiting an infection [11], Wilson et al. did not show a significant association between HbA1c levels, if already elevated, and infection risk [12]. Obesity constitutes an important risk factor for IPP infection overall, with an OR of 18.24 (OR 18.24 95% CI 1.43–231.98) [6]. Specifically for fungal infections, obesity was present in 85% of cases, as reported by Gross et al. [10]. This may not be immediately evident, since an underlying pathology that may be overlooked nowadays is malnutrition [13]. It has become clear that body habitus, i.e., obesity, does not necessarily express a good nutritional feeding status. Often, even obese people may have important nutritional deficiencies compromising their immunity [14]. Another important risk factor contributing to IPP infections is operative time. Every minute that passes operatively increases the chance of an infection [11]. Also, the level of the surgeon’s experience has been shown to play a role in infection risk. It is worth noting that most implants are placed by surgeons who do not perform more than four IPP surgeries in a year [11]. Closed surgical suction drains have not been associated with an increase in infection rates, as shown by a multicenter study [15]. This has been further supported by a Cochrane meta-analysis; however, the study was based on orthopedic surgical suction drains, reducing its applicability to IPP surgery [11]. Other associated risk factors are concomitant surgery, such as circumcision, while urinary sphincter implantation has been associated with increased infection risk in some studies [16]. Contrary to this finding, according to recent literature, there is no significant risk regarding double implantation [6]. As expected, primary IPP surgery showed a reduced risk of infection compared to revision surgery. No difference in infection risk between malleable penile prosthesis and IPP was found. It is worth noting that although the urinary tract is not breached during IPP surgery, urinary tract infections should be treated prior to surgery, since urinary pathogens may enter the operative field. Therefore, according to the AUA guidelines, IPP should not be performed if there is a systemic, cutaneous, or urinary tract infection present. This is because these sites may harbor microorganisms that could spread, making it a fundamental clinical principle [2]. Another major risk factor investigated by Carvajal et al. is immunosuppression [6]. Specifically, the chance of infection increased to 31% in comparison to 1.7% in patients without any immunosuppression (OR 20.99, 95% CI 0.71–622.45). Despite the generally increased risk of infection, none of the patients with fungal infection in the multicenter investigation of Gross et al. were immunosuppressed [10]. Also, no correlation between solid organ transplantation and IPP surgery was identified [6]. Also, environmental factors have been examined. In an interesting study, Gross et al. linked climate conditions to the occurrence of infections in penile prostheses [17]. The study noted that infections were more prevalent in June and less so during winter months, with March having the lowest incidence. Notably, a higher frequency of fungal infections was observed when humidity levels surpassed 55%. This suggests the importance of also considering fungal coverage based on seasonal variations, temperature, and humidity levels [18]. Additionally, it is controversial whether fungal infection of the groin, often in cases of intertrigo, may contribute to increased IPP infections [11]. Lastly, an important and luckily modifiable risk factor is smoking. Smokers are 79% more likely to have an infection compared to non-smokers [11]. Therefore, smoking cessation should always be encouraged [11].

### 2.3. Pathogenesis

In cases where no concurrent infection is present, device infection may begin with the contamination of the IPP with microorganisms before, during, or after surgery. IPP infections occur when bacteria are introduced into the surgical incision during the operation, primarily originating from both the surgical team and the patient’s skin. Although a phenomenon of “a race for the surface,” with fibrous capsule formation of the host macrophages, has been described in a trial to isolate the IPP from the body, at the same time, studies have shown that almost all IPPs are covered by microorganisms, independently of the presence of a clinical infection [19]. Factors such as inadequate skin preparation, improper wound draping, surgeon mistakes, and improperly sterilized instruments can worsen bacterial infiltration. As noted by an analysis of Wilson et al., bacteria cultured in laboratories are usually planktonic bacteria. These bacteria are free-floating and are different from microorganisms that form biofilms [20]. This was further investigated with modern laboratory technologies by Werneburg et al. [21,22]. In their biofilm analysis using next-generation sequencing, shotgun metagenomics were used and yielded biofilm formation in all IPPs independently of concurrent infection [21]. This sheds light on the complex symbiosis of different microbiota and even their potential protection against infections. Additionally, microbiota may change depending on underlying diseases, such as DM, where alterations in their microbiota have been reported [23]. This dysbiosis may explain the increased risk of infection due to fungi in patients with DM. As soon as bacteria successfully attach and form a biofilm, eradication by the host will be unsuccessful. Additionally, lowered metabolic activity and slime formation protect those bacteria and render them resistant to antibiotic therapy [20]. Although bacterial biofilm formation has been extensively investigated, fungal biofilm formation has been less in focus. Biofilm formation has been divided into three developmental steps: the first (early) phase is characterized by yeast cell adherence, the intermediate phase involves the formation of hyphal forms, and the last (maturation) phase is where an increase in matrix material and extracellular polymeric substances (EPS) contribute to the creation of a three-dimensional architecture [24]. It is evident that high glucose levels promote the formation of these biofilms, especially of *Candida parapsilosis*. As already noted, organisms in biofilms behave differently from freely suspended microorganisms, increasing antimicrobial resistance. It has been reported that *Candida albicans* within biofilms may be up to 2000 times more resistant to antifungal treatments [24]. Also, it seems that there has been a shift in the pathogenic types, showing that in the past, before coated implants, the major pathogenic concerns were staphylococcal species, with almost no fungi isolated. Now, more toxic organisms have been found in cultures such as *Escherichia coli*, *Pseudomonas aeruginosa*, *Enterococci*, and an increase in fungi [25]. More specifically, the most common fungi reported in the literature are Candida species, with *Candida albicans* topping the list (62%), followed by *Candida parapsilosis* (23%) and *Candida glabrata* (8%), as reported by Gross et al. [7,10]. Besides environmental microorganism colonization, host defense mechanisms contribute to IPP infection. Usually, Candida species are commensals as long as normal host defenses are present. Previously mentioned risk factors, such as DM, obesity, and immunosuppression, share the underlying common denominator of disruption of host–microbiome symbiosis, allowing Candida species to act as pathogens [24]. Although Werneburg et al. conclude that “non-pathogenic biofilms may inhibit pathogenic infection” [21], it could be useful to further investigate whether non-pathogenic biofilms might actually form a scaffold, allowing for pathogenic bacterial growth, as multispecies biofilms might be the concern after all [26].

### 2.4. Clinical Manifestations—Diagnosis

Clinically, IPP infections can be differentiated into an immediate infection, usually within one month after surgery, or a delayed infection, often occurring several months or even years after implantation [12,20]. In acute, immediate infection, systemic signs are commonly present, such as inflammation at the surgical site or over the IPP, fever, fluctuance around parts of the device, or purulent drainage from the wound. In delayed infection, symptoms are usually more subtle, with signs of a prolonged fixation of the pump to the scrotal wall accompanied by prolonged pain after implant placement [20]. The symptoms and clinical manifestation in a delayed infection are usually local [20]. A small sinus tract may drain serous fluid, and there might be vague pain with inflation of the IPP. These patients have no acute signs of an infection, accompanied by normal laboratory values (white blood cell count and sedimentation rate) [20]. Formerly, the isolated pathogens used to be coagulase-negative *Staphylococci*, particularly *S. epidermidis*, with a recent shift in the diagnosis of *S. aureus*, *Enterococci*, and fungi since the introduction of retardant-coated implants [20]. In 2018, Gross et al. were able to show that patients with fungal infections were also shown to present relatively quickly for treatment after implantation (5.8 months). This was found to be sooner than those with Gram-negative infections (6.96 months) and much later compared to infections due to anaerobes (2.6 months) [27]. After IPP surgery, postoperative pain should gradually subside over a 3–6 week period, and in case of persistence should be evaluated as a sign of delayed infection. Since DM may be accompanied by neuropathic pain, a trial of oral antibiotics may clarify the causation. If pain subsides on the antibiotics and then recurs with their discontinuation, the presence of an infection is likely. Acute fungal infections with catastrophic outcomes have been reported, such as in a recent case by Culha et al., where a *Trichosporon Asahi* fungus was isolated, causing a glans necrosis and leading to penile amputation [20]. Diagnosing fungal infections often requires extra orders in many hospital laboratories, and false-negative culture rates have been reported, challenging the diagnosis in many cases [4]. The biofilm formations that may not allow correct microorganism sampling add difficulty to proper diagnosis. Up to 50% of missing diagnoses, even in invasive disease, have been reported [28]. Therefore, new molecular methodologies including nonculture analysis like β-D-glucan detection and polymerase chain reaction (PCR) should be evaluated as the standard of care to detect fungal disease. In orthopedic device surgery, next-generation sequencing was able to identify microorganisms in 40% of cases when traditional cultures were negative [29]. Additionally, mixed fungal biofilms have been reported, which also remain undiagnosed with classical culturing methods [30]. Mixed fungal biofilms constitute more than one fungus affecting prosthetics; this also has implications for later treatment. As mentioned earlier, in acute infection, the diagnosis is usually based on clinical and laboratory data. In late infections, often presenting with a subclinical course that is difficult to diagnose, imaging modalities can assist in differentiating between superficial and periprosthetic infections [31]. Imaging can be carried out with ultrasound to show fluid collection or edema surrounding the prosthetic components. Computed tomography (CT) is the method of choice regarding the presence of gas in soft tissues, indicating severe infection. Lastly, magnetic resonance imaging (MRI) helps to determine the cause of persistent chronic penile pain and rule out other possible complications such as mechanical malfunction or dislocation. MRI findings of infection are periprosthetic fluid collections and edema. The MRI contrast enhancing of soft tissue at the implant site can indicate chronic infections [31].

### 2.5. Fungal Infections in other Prosthetic Types in Urology

Generally, a recent review by Patel and Gross found no studies on fungal infections associated with artificial urinary sphincters (AUSs), sacral neuromodulators, or suburethral slings [32]. However, the authors described an increased incidence of fungal infections with indwelling catheter, recent antibiotic use (prophylactic or continuous), and diabetic status [32]. Recently, Werneburg et al. analyzed the microbiota of artificial urinary devices where no fungi were detected [22]. Another study where over 80 AUSs were explanted, without any signs of infection, revealed various amounts of bacterial growth but no fungal growth [33]. This is interesting, since both AUSs and IPPs have an antibiotic coating applied on their surface and are manufactured from silicone materials.

### 2.6. Prophylaxis

Despite most of the reported isolated bacteria in IPP infections not being covered based on the antibiotic prophylaxis suggested by the guidelines, infection rates are low [8,20]. The suggested guideline regimen is first- or second-generation cephalosporin or vancomycin in combination with aminoglycoside [2]. Still, due to the detrimental effects of IPP revision surgery, a broad-spectrum antibiotic prophylaxis that includes antifungal drugs should be considered. A recent multicenter study by Gross et al. showed that the regimen suggested by the AUA increased the likelihood of infections. Contrarily, adding antifungal prophylaxis could reduce infections by 92% [34]. Based on their local antibiograms, the authors adopted a perioperative scheme with vancomycin, piperacillin–tazobactam, and fluconazole. Other methods that decrease infection rates include strictly adhering to surgical techniques, avoiding prolonged wound exposure, and minimizing skin contact (i.e., no-touch technique), as stated by the current EAU guidelines. General surgical principles are suggested, such as shaving with clippers, based on a review by Tanner et al. in 2006 [35]. Preparation with chlorhexidine and alcohol is also recommended. The identification and pre-treatment of patients who are colonized with nasal *S. aureus* with mupirocin and chlorhexidine prior to surgery has been shown to reduce surgical site infection from 4.4% to 0.9% and is suggested by the EAU. A perioperative dip using vancomycin and gentamicin, instead of rifampin, showed the most efficacious combination of antibiotics for preventing postoperative infection in IPP in Coloplast Titan (Coloplast Corp, Minneapolis, MN, USA) in a recent review by Towe et al. [36]. The American Medical Systems (AMS, now owned by Boston Scientific) is covered by InhibiZone^®^ (Boston Scientific, Marlborough, MA, USA), a mixture of rifampin and minocycline, applied during the manufacturing process of AMS IPP. Dipping this prosthesis into antibiotics is specifically prohibited since this will remove or reduce the antibiotic coating (Boston Scientific Corporation, AMS 700 Information) [37]. Despite the above dips being effective against bacterial infections, there are no antifungal regimens included either as coatings or dipping agents.

### 2.7. Infection Management and Prognosis

In case of infection, cultures need to be obtained and the treatment guided depending on an antibiogram. The explantation of the infected implant is the cornerstone of the management of established fungal IPP infections. After the explantation of the IPP, thorough debridement of the infected tissue and irrigation with antifungal-enriched solutions is necessary [38]. In the salvage procedures, all components of the device need to be removed, followed by meticulous mechanical lavage with solutions containing diluted betadine, antibiotics, and half-strength hydrogen peroxide [39]. Revision surgery following explantation in order to re-implant a new IPP once the infection has been adequately controlled is feasible. However, the timing of revision surgery should be carefully considered to minimize the risk of recurrent infection and optimize outcomes. For this reason, the Mulcahy protocol has been developed. The Mulcahy protocol is a salvage procedure designed to prevent reinfection in patients undergoing revision IPP surgery. It involves the use of perioperative antibiotics and antifungal agents, meticulous surgical techniques, and postoperative wound care to minimize the risk of recurrent infections [40]. The use of a malleable penile implant at the time of salvage surgery can help minimize corporal fibrosis and loss in penile length [39]. This can also facilitate revision surgery later. Penile traction devices can also be used as an adjunct to preserve penile length [39]. Especially in cases of revision, irrigation should be considered with vancomycin, piperacillin–tazobactam, and amphotericin B, according to Gross et al. [8]. Oral antibiotics should be given for six weeks after revision surgery and depend on culture results. In cases of negative cultures, a combination of trimethoprim–sulfamethoxazole and amoxicillin–clavulanic acid is suggested as a reasonable option. A consensus of Gross et al. did not recommend oral antifungals unless the cultures were positive for Candida species. Prognostically, Gross et al. showed that fungal infections had a lower salvage rate (60%) than both anaerobes (71%) or Gram-positive (64%) organisms [27]. Prolonged oral antibiotics, such as quinolones (for example, ciprofloxacin 500 mg twice a day), are often needed, following discussion with a local infectious clinician. With a suspected fungal infection, using amphotericin B (intravenous 0.5–0.7 mg/kg/day) or fluconazole (oral 400 mg daily) might be appropriate [39].

### 2.8. Antifungal Agents and Resistance

An in vitro study by Mishra et al. examined whether adding antifungal agents into the dipping solution could have an impact on antibiotic efficacy [41]. The incorporation of antifungal solution (amphotericin B) did not diminish antibiotic efficacy and exhibited significant antifungal effects against Candida species without compromising the zone of inhibition of other bacterial pathogens in vitro. This aligns clinically with the multicenter analysis by Barham et al., in which adding antifungal agents reduced overall infection rates [34]. Unfortunately, this is the only clinical study proving that antifungal agents contribute to a clinically significant reduction in IPP infection rates. Which antifungal agents to be used and which route of application (local or systemic) to be implemented remains to be investigated. In this regard, after all, the only data to be extrapolated are from studies performed on prosthetic joint infections. Here, *C. parapsilosis* shows a greater resistance to anidulafungin compared to fluconazole, with fluconazole being effective enough for both *C. albicans* and *C. parapsilosis* [42]. The usual antifungals being used are triazoles (usually fluconazole) and polyene antimycotics (amphotericin B). Still, there are many alternatives, especially in the triazole group, to consider. For example, posaconazole is often used in cases refractory to itraconazole or fluconazole therapy [43]. A major benefit of posaconazole is specifically in difficult-to-treat fungal infections, as in the case of IPP salvage therapies. The therapy is well tolerated, with the most common side effects being gastrointestinal disturbances [43]. Also, posaconazole has been shown to be more effective in invasive disease and immunocompromised patients; however, how this translates to better care in patients undergoing IPP surgery remains to be investigated [44]. Still, approximately 7% of Candida blood samples analyzed at the CDC exhibit resistance to the antifungal medication fluconazole. While *Candida albicans* is the predominant cause of severe Candida infections, resistance is more prevalent among other species, notably *Candida auris*, *Candida glabrata*, and *Candida parapsilosis* [45]. Concern arises over resistance to echinocandins, another class of antifungal drugs. There is a notable trend of increasing echinocandin resistance, particularly in *Candida glabrata*. CDC surveillance data show consistent high resistance levels to fluconazole in *C. glabrata* over two decades [45]. Generally, there has been noted an emerging resistance overall for Candida species to echinocandins (i.e., caspofungin). Patients with Candida infections resistant to both fluconazole and echinocandins have limited treatment options. The main therapy in those cases is amphotericin B, which can pose toxicity risks, especially in patients with comorbidities [45] (Table 1).

### 2.9. IPP Materials

It is evident that hydrophilic coatings of AMS InhibiZone and Coloplast on IPP have reduced device infections [19]. Different fungi show differences in hydrophilicity, with *C. albicans* being more hydrophilic than *C. parapsilosis* [42]. Incorporating phosphorous compounds might enhance cellular proliferation, potentially counteracting any cytotoxic effects of antifungal agents. For penile implants, this highlights the importance of biocompatibility in coating design [42]. Reasonable coatings could be silver nanoparticles (AgNPs) due to their antibacterial and antifungal properties [46]. AgNPs can disrupt bacterial and fungal cell membranes, leading to cell death. Also, AgNPs, when properly synthesized and coated, have shown minimal cytotoxicity and tissue compatibility, making them suitable for use in medical implants, such as surface coatings for IPP. Additionally, unlike some conventional antibiotics, silver nanoparticles have a lower likelihood of inducing microbial resistance. This is advantageous in long-term implants like penile inflatable implants, where prolonged exposure to microbial flora is expected [46]. Other advancements in nanofabrication to tackle biofilms have been developed, such as surface nanotopographies, which can reduce microbial adhesion [47]. Also, developments in superhydrophobic materials promise microbial repulsion and even bactericidal activity. However, changes in topography may prohibit the development of certain microorganisms, but at the same time offer better environmental conditions for other microorganisms. For example, hydrophobic surfaces may just reduce the chance of biofilm development of hydrophilic microorganisms but help hydrophobic microorganisms to form biofilms (Table 2).

## 3. Future Directions

Future research is directed towards recognizing and further investigating the molecular mechanisms of pathogenic or non-pathogenic biofilm formation, such as their interplay and cross-talking molecules. Furthermore, an in-depth understanding of the above mechanisms will consecutively allow the development of biomaterials with either biofilm-formation-inhibiting capacity or even non-pathogenic biofilm coatings to help reduce the infection rate. Also, the quest for new IPP materials will be accelerated by artificial intelligence-driven machine learning algorithms [48]. Final thoughts: addressing lifestyle factors, promoting education about ED risk factors, and offering psychological support may reduce ED and, thereby, IPP surgery needs. Research into non-invasive treatments is vital. Public health strategies target ED’s root causes, alleviating its burden.

## 4. Conclusions

This review underlines the added benefit and importance of antifungal treatment, not only during salvage, but also in virgin IPP cases. Although not concurrent with international guidelines, the evidence in the medical literature has been provided, and given the detrimental effect of fungal IPP infection on patient satisfaction and life, it is reasonable to implement antifungal prophylaxis in patients with risk factors (DM, obesity, immunosuppression). Still, standard practices, such as the no-touch technique, continuation of antibiotic therapy, preparation with chlorhexidine and alcohol, shaving, glycemic control, and antibiotic dips, are the cornerstone of IPP infection prophylaxis.

## Figures and Tables

**Table 1 jpm-14-00644-t001:** Recommended prophylaxis and management agents, along with their dosages.

Drug	Dosage	Indication	Comments
Fluconazole	400 mg orally, 1 h before surgery, then 200 mg daily for 14 days	Prophylaxis	Effective against most Candida species
Liposomal Amphotericin B	3–4 mg/kg IV daily	Treatment of established infections	Reserved for severe infections due to toxicity
Caspofungin	70 mg IV on day 1, then 50 mg IV daily	Salvage therapy for invasive infections	Well-tolerated with fewer side effects compared to Amphotericin B
Piperacillin/Tazobactam	3.375 g IV every 6 h	Broad-spectrum antibacterial coverage	Useful in polymicrobial infections
Vancomycin	15–20 mg/kg IV every 8–12 h	Prophylaxis and treatment of Gram-positive infections	Monitor levels to avoid toxicity
Ciprofloxacin	400 mg IV every 12 h or 500 mg orally twice daily	Broad-spectrum antibacterial coverage	Effective against Gram-negative and some Gram-positive bacteria
**AUA Guideline Regimen**	1st- or 2nd-generation cephalosporin or vancomycin + aminoglycoside	Standard prophylaxis	Combination provides broad-spectrum coverage

**Table 2 jpm-14-00644-t002:** A compilation of currently available IPP materials.

Device	Manufacturer	Company Office	Material	Characteristics
AMS 700 Series	American Medical Systems (AMS)	Minnetonka, MN, USA	Silicone	Inflatable, high biocompatibility, InhibiZone antibiotic coating available.
AMS Ambicor	American Medical Systems (AMS)	Minnetonka, MN, USA	Silicone	Two-piece inflatable, easy to use, suitable for patients with limited dexterity.
AMS Spectra	American Medical Systems (AMS)	Minnetonka, MN, USA	Silicone	Malleable, bendable rods for a constant state of erection, easy to position.
Coloplast Titan	Coloplast	Humlebaek, Denmark	Bioflex	Inflatable, enhanced durability, reduced infection rates.
Coloplast Genesis	Coloplast	Humlebaek, Denmark	Silicone	Malleable, reliable, easy to implant.
Rigicon Infla10	Rigicon	Ronkonkoma, NY, USA	Hydrophilic Coating	Three-piece inflatable, better tissue integration.
Rigicon Rigi10	Rigicon	Ronkonkoma, NY, USA	Customizable Silicone	Malleable, customizable for patient-specific needs.
Zephyr Implant	Zephyr Surgical Implants	Geneva, Switzerland	Soft Silicone	Customizable, innovative design under investigation.

## Data Availability

No new data were created or analyzed in this study. Data sharing is not applicable to this article.

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
