# Peer review of "Unveiling the Hidden Perils: A Comprehensive Review of Fungal Infections in Inflatable Penile Prosthesis Surgery"

_jpm, 2024, doi:10.3390/jpm14060644_

Round 1

Reviewer 1 Report

Comments and Suggestions for Authors

In section 2.1 "Epidemiology" the "5%" that are fungal infections should be qualified to be 5% of the infectious complications so as not to confuse readers after saying infectious complications occur in "1-3%" in the previous line

Author Response

Patras, June 1, 2024

Dear Editor,

I would like to thank you and the reviewers for evaluating our manuscript entitled ‘Unveiling the Hidden Perils: A Comprehensive Review of Fungal Infections in Inflatable Penile Prosthesis Surgery.

We have followed the reviewers’ recommendations and remarks and hope to meet them adequately as you can see in the point-to-point rebuttal letter. To facilitate the re-review process, we have applied line numbering throughout the revised manuscript. Omissions are seen in red, whereas changes and additions are highlighted in yellow in the revised manuscript.

We are looking forward to hearing from you.

Sincerely,

According to comments raised by Reviewer 1:

 In section 2.1 "Epidemiology" the "5%" that are fungal infections should be qualified to be 5% of the infectious complications so as not to confuse readers after saying infectious complications occur in "1-3%" in the previous line

Authors’ Response:

Thank you for your valuable feedback. We appreciate your careful review and suggestions for clarity. We have revised section 2.1 "Epidemiology" to specify that the "5%" refers to fungal infections as a percentage of infectious complications. The revised sentence now reads: " Studies have reported that fungal infections account for approximately 5% of all infections ..." This should help prevent any confusion among readers.

Reviewer 2 Report

Comments and Suggestions for Authors

This scientific work is very interesting, considering the relatively small number of these types of interventions or their predilection for specialized centers.

The data presented is well structured, no bias or major linguistic errors are noted, but there are some essential aspects that should be taken as a message in one form or another:

1. as the authors have stated, the experience of the surgeon is crucial - so we are talking about specialized centers / that frequently perform this type of procedure

2. the guidelines are strict guidelines, they cannot justify the omission of prophylaxis as long as it is considered necessary in the patient's case

Author Response

Patras, June 1, 2024

Dear Editor,

I would like to thank you and the reviewers for evaluating our manuscript entitled ‘Unveiling the Hidden Perils: A Comprehensive Review of Fungal Infections in Inflatable Penile Prosthesis Surgery.

We have followed the reviewers’ recommendations and remarks and hope to meet them adequately as you can see in the point-to-point rebuttal letter. To facilitate the re-review process, we have applied line numbering throughout the revised manuscript. Omissions are seen in red, whereas changes and additions are highlighted in yellow in the revised manuscript.

We are looking forward to hearing from you.

Sincerely,

According to comments raised by Reviewer 2:

 This scientific work is very interesting, considering the relatively small number of these types of interventions or their predilection for specialized centers.

The data presented is well structured, no bias or major linguistic errors are noted, but there are some essential aspects that should be taken as a message in one form or another:

  1. as the authors have stated, the experience of the surgeon is crucial - so we are talking about specialized centers / that frequently perform this type of procedure
  2. the guidelines are strict guidelines, they cannot justify the omission of prophylaxis as long as it is considered necessary in the patient's case

Authors’ Response:

We are grateful for your positive comments and for highlighting important aspects of our work.

  1. We agree with your observations which are indeed emphasized in the manuscript.

2. While guidelines are strict, we do not support an omission of prophylaxis and we stress the importance of adhering to prophylactic measures based on individual patient risk assessments and extending guidelines prophylaxis.

Reviewer 3 Report

Comments and Suggestions for Authors

I have read the manuscript titled "Unveiling the Hidden Perils: A Comprehensive Review of Fungal Infections in Inflatable Penile Prosthesis Surgery." The authors aimed to investigate the fungal complications associated with IPP surgery. The manuscript appears well-structured and articulated, with conclusions supported by the current literature. However, the authors may wish to revise the manuscript considering the following comments.

1. Some typesetting modifications are needed. For instance, in line 142, "Candida albicans" should be italicized. Please review the entire manuscript to rectify similar issues.

2. The target audience defined by the authors is urologists; however, there are no figures or tables to facilitate readability. For instance, incorporating tables that elaborate on the recommended prophylaxis and management agents, along with their dosages, would be of interest to urologists. Additionally, compiling tables related to IPP materials currently available and under investigation is recommended.

Author Response

Patras, June 1, 2024

Dear Editor,

I would like to thank you and the reviewers for evaluating our manuscript entitled ‘Unveiling the Hidden Perils: A Comprehensive Review of Fungal Infections in Inflatable Penile Prosthesis Surgery.

We have followed the reviewers’ recommendations and remarks and hope to meet them adequately as you can see in the point-to-point rebuttal letter. To facilitate the re-review process, we have applied line numbering throughout the revised manuscript. Omissions are seen in red, whereas changes and additions are highlighted in yellow in the revised manuscript.

We are looking forward to hearing from you.

Sincerely,

According to comments raised by Reviewer 3:

 I have read the manuscript titled "Unveiling the Hidden Perils: A Comprehensive Review of Fungal Infections in Inflatable Penile Prosthesis Surgery." The authors aimed to investigate the fungal complications associated with IPP surgery. The manuscript appears well-structured and articulated, with conclusions supported by the current literature. However, the authors may wish to revise the manuscript considering the following comments.

  1. Some typesetting modifications are needed. For instance, in line 142, "Candida albicans" should be italicized. Please review the entire manuscript to rectify similar issues.
  2. The target audience defined by the authors is urologists; however, there are no figures or tables to facilitate readability. For instance, incorporating tables that elaborate on the recommended prophylaxis and management agents, along with their dosages, would be of interest to urologists. Additionally, compiling tables related to IPP materials currently available and under investigation is recommended.

Authors’ Response:

Thank you for your thorough review and constructive suggestions. We have made the following revisions based on your feedback:

  1. We have reviewed the entire manuscript for typesetting modifications and corrected all instances where species names, such as "Candida albicans," should be italicized.
  2. To enhance readability and provide valuable information for our target audience of urologists, we have included several tables. These tables include:

o                Recommended prophylaxis and management agents, along with their dosages.

o                A compilation of IPP materials currently available .

We believe these additions will greatly benefit our readers and improve the overall utility of our manuscript.

Round 2

Reviewer 3 Report

Comments and Suggestions for Authors

I have now assessed the revised manuscript and found that the authors have addressed all of my previous concerns. The overall quality of the manuscript has been improved, and the revised manuscript can now be accepted in its current form.